# Revisiting the Checkerboard to Inform Development of β-Lactam/β-Lactamase Inhibitor Combinations

**DOI:** 10.3390/antibiotics13040337

**Published:** 2024-04-07

**Authors:** Darren J. Bentley

**Affiliations:** Certara Drug Development Solutions, Certara Level 2-Acero, 1 Concourse Way, Sheffield S1 2BJ, UK; darren.bentley@certara.com; Tel.: +44-(0)7776-529384

**Keywords:** antibiotic, beta-lactam, beta-lactamase, combination, checkerboard, interaction, synergy, MIC, pharmacokinetic/pharmacodynamic

## Abstract

A two-dimensional “checkerboard” array employing systematic titration (e.g., serial two-fold dilutions) is a well-established in vitro method for exploring the antibacterial effects of novel drug combinations. Minimum inhibitory concentrations (MICs) on the checkerboard are isoeffective points at which the antibiotic potency is the same. Representations of checkerboard MIC curves for a β-lactam and β-lactamase inhibitor combination are used in hypothetical “thought experiments” and reveal the ways in which current practices can be improved. Because different types of response (i.e., independence vs. additivity vs. one effective agent; interaction vs. noninteraction) produce different MIC curves, data from different strains/isolates should not be pooled indiscriminately, as the composition of a pooled dataset will influence any derived pharmacokinetic/pharmacodynamic (PK/PD) index. Because the β-lactamase inhibitor threshold concentration (C_T_) parameter is a function of the β-lactam partner dosing regimen, it is not possible to derive a universal PK/PD index target based on C_T_. Alternative susceptibility testing methods represent different planes through the checkerboard; a fixed ratio method is less prone to bias for all β-lactam and β-lactamase inhibitor combinations. Susceptibility test MICs will often not reflect the sensitivity of the strain/isolate to the β-lactamase inhibitor, so the use of these MICs to normalize PK/PD indices is inappropriate.

## 1. Introduction

The global public health threats from antimicrobial resistance and, in particular, the emergence of multidrug-resistant Gram-negative bacterial infections have spurred efforts to develop new β-lactamase inhibitors to counteract resistance to established β-lactam antibiotics [1,2,3,4,5,6,7]. The β-lactamase inhibitor prevents inactivation of the β-lactam and, thereby, re-sensitizes bacteria to the killing effects of the β-lactam. In addition, some β-lactamase inhibitors (e.g., durlobactam, nacubactam, and zidebactam) also have direct antibacterial activity linked to inhibition of penicillin-binding proteins (PBPs) [4]. A number of β-lactam plus β-lactamase inhibitor combinations have received regulatory approval for clinical use, and more are currently in development [8]. All programs rely extensively on established pharmacokinetic/pharmacodynamic (PK/PD) principles to extrapolate nonclinical information and guide clinical development decisions [9,10,11,12,13], but there remains a lack of consensus about the best way to optimize dosing of the β-lactamase inhibitor (e.g., [14]), particularly if the β-lactamase inhibitor also has direct antibacterial activity.

A two-dimensional “checkerboard” array employing systematic titration (e.g., serial two-fold dilutions) of two drugs is a well-established in vitro method for exploring the antibacterial effects of novel antibiotic combinations. Minimum inhibitory concentrations (MICs) on the checkerboard represent isoeffective points (i.e., antibiotic potency is the same at all points identified as MICs), and the shape of a checkerboard MIC curve reflects sensitivities to each drug and any “interactions” among drugs. Representations of checkerboard MIC curves have been used in “thought experiments” for a hypothetical combination of a β-lactam plus a β-lactamase inhibitor, both with and without direct antibacterial activity.

## 2. Results

### 2.1. Representation of Checkerboard Data and Combination Response Models

At its simplest, a “checkerboard” analysis is a systematic titration of the concentrations of two drugs and a measurement of the effects on bacterial growth by each concentration pair. While it is possible to derive quantitative estimates of bacterial growth, in practice the most common approach is to use a growth/no growth binary endpoint to determine minimum inhibitory concentrations (MICs). The MICs represent the lowest concentration combinations that prevent visible growth of bacteria. Hence, the MICs on a checkerboard denote points at which antibacterial potency is the same (i.e., isoeffective), and, thus, the MIC curve is a response surface connecting isoeffective points (i.e., isobole).

The shape of a checkerboard MIC curve will reflect the sensitivity of the test strain/isolate to each component and the nature of any interaction between it and the components. A number of different factors can, therefore, influence the shape of the MIC curve (see Section 2.2) and, consequently, every bacterial strain/isolate could potentially have its own almost unique MIC curve. Nevertheless, representative MIC curves can be simulated using generalized combination response models, which have been developed in order to quantitatively assess the synergy/antagonism of drug combinations. These methods are described in detail elsewhere (e.g., [15,16,17,18,19,20,21]), and the most commonly used models for β-lactam and β-lactamase inhibitor combinations include Bliss independence, Loewe additivity, indifference (highest single agent) and inertism. The simulation of drug effect isoboles for each of these response models allows for the representation of hypothetical MIC curves on a checkerboard under scenarios assuming either no interaction or synergistic interactions between the components of a β-lactam and β-lactamase inhibitor combination (Figure 1). Combinations of a β-lactam with a conventional β-lactamase inhibitor with minimal or no direct antibacterial effect are described by an inertism model; combinations of a β-lactam with a newer β-lactamase inhibitor that also has direct antibacterial activity when used as monotherapy can potentially be described by Bliss independence, Loewe additivity, or highest single agent models.

The hypothetical MIC curves for a β-lactam plus β-lactamase inhibitor combination (Figure 1) illustrate that each combination response model produces distinctive MIC curves on a checkerboard. In principle checkerboard data can be used to evaluate the predominant mechanism of action and type of interaction occurring between β-lactam and β-lactamase inhibitor components in each bacterial strain/isolate. For example, it is possible to visually identify strains/isolates in which a β-lactamase inhibitor has appreciable direct antibiotic effects (i.e., MIC curve intersects the X axis) and those in which the β-lactamase inhibitor only acts to potentiate the effects of the β-lactam partner, either by inhibition of β-lactamase activity, or through an “enhancer” effect of undetermined mechanism (e.g., [22,23]), i.e., the MIC curve does not intersect the X axis. In practice, while MIC curves tend to have shapes that are less regular than the theoretical isoboles, the curves from most strains/isolates will fit a recognizable pattern (personal observation). In particular, it is possible to identify those strains/isolates for which there is no evidence of meaningful direct antibacterial activity from the β-lactamase inhibitor. Hence, it is possible to use a visual examination of checkerboard data to group strains/isolates according to the predominant mechanism of action when attempting to determine PK/PD indices and targets. However, it is not possible to use checkerboard data in isolation to unambiguously differentiate among models in scenarios in which the β-lactamase inhibitor has direct antibacterial effects. As previously noted in [24] and illustrated in Figure 1, the combination response models show degeneracy, i.e., two or more models can describe similar MIC curves. Therefore, identification of the most appropriate combination response model for synergy assessments requires additional data about the mechanistic basis for interactions between β-lactam and β-lactamase inhibitor components.

### 2.2. β-Lactamase Inhibitor Dose Selection

The sensitivity of a strain/isolate to a β-lactamase inhibitor (i.e., the β-lactamase inhibitor concentration range within which an MIC is altered) will be influenced by a number of factors, including affinity of the β-lactamase inhibitor for the β-lactamase enzyme and PBP molecules (e.g., inhibitory constants (K_i_s)), intrinsic β-lactamase enzyme activity (i.e., rate of reaction catalyzed by one molecule of β-lactamase), β-lactamase enzyme and PBP expression (i.e., number of β-lactamase and PBP molecules per cell), and the β-lactamase inhibitor concentration at the intracellular target sites (i.e., cellular influx/efflux transport of the β-lactamase inhibitor). Only the affinity of the β-lactamase inhibitor for the β-lactamase enzyme and PBP molecules is a characteristic of the β-lactamase inhibitor; all other factors are characteristics of the pathogen being treated. Each of these factors will alter the MIC curves for a β-lactam plus β-lactamase inhibitor combination in different ways (see Figure 2 as an example for a β-lactamase inhibitor without direct antibacterial activity).

The identification of a pharmacologically active concentration range for a β-lactamase inhibitor is possible by examination of checkerboard data to determine the inflection points of MIC curves for individual strains/isolates. However, because the factors affecting β-lactamase and PBP molecular targets are distinct (e.g., K_i_s vs. each target are different), sensitivity to a β-lactamase inhibitor will potentially diverge between those strains/isolates for which the inhibition of β-lactamase activity is the principal mechanism of action (i.e., one effective agent scenarios; Figure 1c,d,g,h) and those strains/isolates for which the inhibition of PBP activity is the principal mechanism of action (i.e., independence and additivity scenarios; Figure 1a,b,e,f). Therefore, when attempting to identify a pharmacologically active concentration range for a β-lactamase inhibitor using in vitro or in vivo data, consideration should be given to which mechanism of action predominates when pooling data across strains/isolates.

When attempting to establish pharmacokinetic/pharmacodynamic indices and targets for β-lactamase inhibitors without direct antibacterial activity, it is established practice to attempt to identify a threshold concentration (C_T_) for the β-lactamase inhibitor, and derive the fraction of the dosing interval that β-lactamase inhibitor concentrations are maintained above that threshold concentration (%T > C_T_ or unbound (“free”) f%T > C_T_) [10]. C_T_ represents the lowest β-lactamase inhibitor concentration that produces a pharmacologically meaningful inhibition of β-lactamase activity, i.e., the lowest concentration that causes a detectable shift in the MIC on a checkerboard. However, checkerboard MIC curves demonstrate that C_T_ for a β-lactamase inhibitor is dependent on the concentration of the β-lactam partner. Hence, it is appropriate to differentiate “absolute” C_T_, representing the minimum β-lactamase inhibitor concentration that produces a shift in the MIC, and “relative” C_T_, representing the minimum β-lactamase inhibitor concentration that produces shift in the MIC at a given β-lactam concentration (Figure 3). For an individual MIC curve, the absolute and relative C_T_ values may be similar if the MIC curve is steep (i.e., approximating to a step function), but they will be different where the MIC curve is shallow. The absolute C_T_ will always be lower than or equal to the relative C_T_ and will only be identifiable in practice using fixed concentrations of the β-lactam partner (e.g., [25]). In dynamic in vitro and in vivo model systems in which β-lactam concentrations change with time, while the absolute C_T_ will remain constant, the relative C_T_ will fluctuate. Therefore, while the absolute C_T_ is a function of the strain/isolate being treated, the relative C_T_ is also a function of the β-lactam partner and β-lactam dosing regimen. Hence, any empirically derived %T > C_T_ parameter based on a single fixed C_T_ value will be dependent on—and specific for—the β-lactam partner and dosing regimen.

### 2.3. Drug Susceptibility Testing Methods

Standardized methods for antibiotic susceptibility testing are set out in guidelines by organizations such as the European Committee on Antimicrobial Susceptibility Testing (EUCAST) and Clinical and Laboratory Standards Institute (CLSI). All testing methods are based on the determination of MICs, which means that it is possible to represent standard susceptibility testing methods in the same way as checkerboard data and combination response models (see Section 4). For β-lactam plus β-lactamase inhibitor combinations, two alternative susceptibility testing strategies are in use, as follows: a “fixed concentration” approach in which concentrations of the β-lactam are titrated, while the concentration of the β-lactamase inhibitor remains constant; or a “fixed ratio” approach in which concentrations of the β-lactam and β-lactamase inhibitors are both titrated in unison so that the relative ratio between the two remains constant. As illustrated in Section 4, the alternative susceptibility testing methods represent different planes or “slices” through the checkerboard, and, hence, the MIC value estimated by each method may differ. It is apparent the MIC values reported will depend on the experimental β-lactamase inhibitor concentration or β-lactam:β-lactamase inhibitors ratio chosen, as well as the shape of the true MIC curve. It is, therefore, not possible to directly compare MIC values derived from the different methodologies, and—for reasons discussed further below—reported “fixed-concentration” MIC values will, generally, tend to be lower than reported “fixed-ratio” MIC values in most scenarios.

#### 2.3.1. “Fixed Concentration” Method

The representation of different β-lactamase inhibitor concentrations in a “fixed concentration” testing method as vertical planes through a hypothetical checkerboard MIC curve (Figure 4) illustrates that different fixed β-lactamase inhibitor concentrations will potentially produce markedly different reported MIC values. The reported MIC values will be inversely related to the fixed β-lactamase inhibitor concentration employed (i.e., MIC values will tend to be lower with higher β-lactamase inhibitor concentrations). Depending on the shape of the MIC curve, it is possible for the reported MIC value to change dramatically over a limited β-lactamase inhibitor concentration range (i.e., one log2 increment), implying that reported MIC values are potentially highly sensitive to the choice of β-lactamase inhibitor concentration. In scenarios in which the β-lactamase inhibitor has no direct antibacterial activity (i.e., a monotherapy MIC value cannot be identified), the reported MIC from a combination will reach an identifiable minimum value as the fixed β-lactamase inhibitor concentration increases. However, in scenarios in which the β-lactamase inhibitor has antibacterial activity (i.e., monotherapy MIC values can be identified for both β-lactam and β-lactamase inhibitor components), an arbitrary vertical plane may not bisect the true MIC curve at all. In such circumstances, the reported MIC value at higher β-lactamase inhibitor concentrations will be below the lowest β-lactam concentration tested (i.e., effectively zero) and will not have any quantitative meaning.

It is accepted practice that “fixed concentration” susceptibility testing methods employ a concentration of the β-lactamase inhibitor that has been empirically established to have the largest effect in vitro (e.g., [26,27]). Hence, the reported MIC value represents sensitivity to the β-lactam component of the combination in the presence of a maximally pharmacologically active concentration of the β-lactamase inhibitor and will, thus, represent the lowest achievable MIC for the combination (Figure 4). However, the reported MIC value will correlate poorly with the sensitivity of the strain/isolate to the pharmacological effects of the β-lactamase inhibitor, i.e., strains/isolates with the same reported MIC values can have very different sensitivities to the β-lactamase inhibitor component while strains/isolates with similar sensitivity to the β-lactamase inhibitor component may have very different reported MIC values.

#### 2.3.2. “Fixed Ratio” Method

The representation of different β-lactam:β-lactamase inhibitor ratios in a “fixed ratio” testing method as diagonal planes through a hypothetical checkerboard MIC curve (Figure 5) illustrates that different fixed ratios will lead to different reported MIC values. The reported MIC values will be inversely related to the fixed ratios employed (i.e., MIC values will tend to be lower when the relative proportion of the β-lactamase inhibitor is higher). For any MIC curve the reported MIC value will only differ by a maximum of one log2 value for any single increment change in the fixed ratio (e.g., 1:1 to 1:2). In scenarios in which the β-lactamase inhibitor has no direct antibacterial activity (i.e., a monotherapy MIC value cannot be identified), the reported MIC from a combination will reach an identifiable minimum value as the fixed ratio increases. Where the β-lactamase inhibitor has direct antibacterial activity, the reported apparent MIC will continue to decrease as the relative proportion of the β-lactamase inhibitor in the fixed ratio increases. However, in all in scenarios, regardless of whether the β-lactamase inhibitor has antibacterial activity, any arbitrary diagonal plane will always bisect the true MIC curve at some point, and, thus, the reported MIC values will always be nonzero.

The reported MIC from a fixed ratio testing method will be influenced by the sensitivity to both β-lactam and β-lactamase inhibitor components. As discussed in Section 2.2 (“β-Lactamase Inhibitor Dose Selection”), this will reflect the influences of a number of factors that affect the shape of the MIC curve (Figure 2). Considering a 1:1 fixed ratio, the reported MIC of the combination will be equal to or below the lowest monotherapy MIC. Consequently, in situations in which the β-lactam monotherapy MIC is lower than the β-lactamase monotherapy MIC, the reported MIC value for the combination will not directly reflect the sensitivity of the strain/isolate to the pharmacological effects of the β-lactamase inhibitor.

## 3. Discussion

Thought experiments employing hypothetical checkerboard MIC curves offer insights which can inform the development of β-lactam plus β-lactamase inhibitor combinations and, in particular, guide strategies to select the β-lactamase inhibitor dose. The shape of a checkerboard MIC curve will reflect the bacterial sensitivity to each component of the combination, and the nature of any interaction among the components. Although the shape of MIC curves may be complex if multiple mechanisms are operating simultaneously, experience suggests that curves from most strains/isolates will fit a recognizable pattern. Hence, visual examination of MIC curves can be used to evaluate the predominant mechanism of action in each bacterial strain/isolate and allows them to be grouped accordingly.

The categorization of strains/isolates based on the predominant mechanism of action is particularly important when attempting to establish PK/PD indices for β-lactamase inhibitors with direct antibacterial effects, as well as inhibiting β-lactamase activity. While they have limitations (e.g., [28]), PK/PD indices are extensively used to extrapolate results from nonclinical models to a clinical setting and support clinical dose selection [9,10,11,12,13,29,30,31]. Standard PK/PD indices are believed to reflect the underlying mechanism of antibacterial activity, e.g., C_max_/MIC is associated with concentration-dependent bactericidal activity, while %T > MIC is associated with time-dependent effects. Therefore, in individual strains/isolates for which different mechanisms of action predominate and drug effects are mediated by different targets (i.e., β-lactamase vs. PBP), the corresponding PK/PD indices may be different. Similarly, the pharmacologically relevant β-lactamase inhibitor concentration range (i.e., concentration range within which the MIC changes) may also be different. The implication is that any PK/PD index or target value empirically derived using pooled data from a heterogeneous mix of strains/isolates will be influenced by the composition of the dataset. At best this will introduce variability, which will confound the analysis, and at worst could result in derivation of a PK/PD index parameter and target value that cannot be extrapolated to other datasets or settings. The best practice is, therefore, to derive the PK/PD index parameter and target values for each strain/isolate individually or to use checkerboard information to selectively pool strains/isolates according to the predominant mechanism of action.

The existence of distinct mechanisms of action for β-lactamase inhibitors with direct antibacterial effects also highlights a weakness in the common practice of applying fractional inhibitory concentration (FIC) analysis to assess synergy. There are a variety of methods to assess synergy—and rule out antagonism—for drug combinations by evaluating the differences between observed responses and expected responses from a null reference model assuming no interaction or additivity (e.g., [32,33,34]). A review of these methods is beyond the scope of this manuscript, but, while there is no consensus on the best approach, FIC analysis is commonly used on checkerboard data in scenarios in which the β-lactamase inhibitor has direct antibacterial effects. An FIC analysis is a nonparametric method based on Loewe additivity zero interaction theory, which assumes mutual exclusivity, i.e., that both drugs have the same target [35,36]. While this assumption may be valid for some β-lactam and β-lactamase inhibitor combinations for which the penicillin-binding protein (PBP) binding profiles overlap [37], it does not hold for all β-lactam and β-lactamase inhibitor combinations. Universal application of an FIC analysis is, therefore, inappropriate and potentially misleading, and an evaluation of synergy in checkerboard data requires critical consideration of pharmacological mechanisms in order to select an analysis method that uses the most appropriate null reference model.

Despite their limitations, PK/PD indices incorporating susceptibility test MIC values (e.g., C_max_/MIC, AUC/MIC, and %T > MIC) are extensively used in antibiotic drug development, and regulatory guidelines recommend their use for β-lactamase inhibitor dose selection [14]. The purpose of including the MIC term is to correct for differences in sensitivity to drug among organisms and thereby derive a single parameter that can be generalized across all strains/isolates. However, the observation that in many scenarios the reported MIC value will not directly reflect the sensitivity of the strain/isolate to the pharmacological effects of the β-lactamase inhibitor indicates that this approach is flawed for the purposes of β-lactamase inhibitor dose selection. The use of standard susceptibility test MICs to normalize β-lactamase inhibitor exposure parameters introduces an unacknowledged source of error into analyses. Similarly, inclusion of susceptibility test MICs as a parameter in mechanistic or semi-mechanistic PK/PD models (e.g., [38]) will potentially confound model development. A more appropriate approach for β-lactamase inhibitor dose selection is to relate β-lactamase inhibitor exposure to C_T_ as a measure of β-lactamase inhibitor sensitivity (e.g., %T > C_T_), as exemplified by the development strategy adopted for avibactam [39].

The pharmacologically relevant concentration range for the β-lactamase inhibitor in vitro can be identified by the shift in the MIC on a checkerboard, but the minimum β-lactamase inhibitor concentration that produces the shift in the MIC is dependent on the β-lactam concentration. Hence, a C_T_ value will be a function of the β-lactam partner and dosing regimen, as well as the pathogen being treated. In turn this implies that if using an invariant C_T_ value it will not be possible to derive a single %T > C_T_ pharmacokinetic/pharmacodynamic index target that is common to all β-lactam partners and dosing regimens, and that for the selection of a clinical β-lactamase inhibitor dosing regimen, priority should be given to in vitro and in vivo models that mimic the clinical dosing regimen of the intended β-lactam partner.

An alternative PK/PD index approach that is increasingly being used is based on “instantaneous” or “dynamic” MICs derived from checkerboard data [40]. In brief, a mathematical model (e.g., E_max_ model) is fitted to checkerboard data to describe the relationship between β-lactamase inhibitor concentration and β-lactam MICs, and this is used to predict “instantaneous” MIC (MIC_i_) values for the β-lactam in dynamic in vitro and in vivo models based on the concentration vs. time profile of the β-lactamase inhibitor [41,42,43,44]. These MIC_i_ values are then used to derive MIC-normalized PK/PD indices for the β-lactam (i.e., %T > MIC_i_). This method is attractive as it offers the potential to derive a single universal PK/PD index for a combination, and it represents a possible alternative to traditional MIC- or C_T_-based indices for the purposes of dose selection. However, it also comes with similar caveats concerning the pooling of data from strains/isolates with different predominant mechanisms of action. In particular, it makes the implicit assumption that all points on a checkerboard isobole (i.e., MIC curve) are functionally equivalent. This is a reasonable assumption for combinations involving β-lactamase inhibitors without direct antibacterial effects, whereby bacterial killing is mediated entirely through the β-lactam partner, but it is less credible for combinations involving β-lactamase inhibitors with direct antibacterial activity. In extremis, it implies that the in vivo antibacterial effects of supra-MIC concentrations of β-lactam monotherapy and β-lactamase inhibitor monotherapy are similar. However, this has not been convincingly demonstrated. For example, it has been reported that even high doses of nacubactam monotherapy are not effective in in vivo models against some carbapenem-resistant Enterobacterales isolates despite low in vitro nacubactam MICs [45]. In such circumstances, mechanistic or semi-mechanistic PK/PD modeling (e.g., [13,31,46,47,48]) offers potential advantages over traditional PK/PD indices.

Although not without limitations (e.g., [49]), in vitro susceptibility testing is a practically convenient way to quantify the sensitivity of an organism to the effects of a drug—or drug combination—in order to differentiate between resistant and susceptible bacteria and guide clinical treatment decisions. For clinical decision making, the avoidance of “false positives” (i.e., test result indicates a resistant organism is susceptible) is more important than “false negatives” (i.e., test result indicates a susceptible organism is resistant). Hence, the susceptibility testing method chosen should minimize the potential bias. Fixed concentration and fixed ratio susceptibility testing methods represent different planes or “slices” through the checkerboard MIC curve to derive a single MIC value. For any strain/isolate, the reported MIC value from standard susceptibility testing will depend on the testing method (i.e., fixed concentration vs. fixed ratio) and conditions (i.e., concentration or ratio value) chosen, as well as the shape of the “true” MIC curve. This means that if using established susceptibility test interpretative criteria (i.e., “breakpoints”) for the β-lactam component the cumulative fractional response and apparent coverage (i.e., proportion of the bacterial population deemed susceptible to treatment) will be determined by the testing conditions chosen. In particular, because reported MIC values will be inversely related to the fixed β-lactamase inhibitor concentrations or fixed ratios employed, coverage will appear greater when the β-lactamase inhibitor concentration or relative proportion is higher. While it is a common—but not universal—practice that susceptibility testing methods employ concentrations or ratios that are clinically relevant (e.g., the fixed ratio matches the ratio at which the β-lactam and β-lactamase components are dosed to patients in vivo [50]), the choice of is essentially arbitrary. Therefore, the determination of susceptibility test interpretative criteria for a β-lactam plus β-lactamase inhibitor combination needs to be supported by data relating clinical treatment outcome to susceptibility test MIC values rather than relying on criteria previously established for β-lactam monotherapy.

Fixed concentration testing methods are more affected by the choice of testing conditions than fixed ratio methods. For a fixed β-lactamase inhibitor concentration, it is possible for the reported MIC value to change dramatically over a limited β-lactamase inhibitor concentration range (i.e., one log2 increment). Furthermore, in scenarios in which the β-lactamase inhibitor has antibacterial activity, the reported MIC value may be effectively zero. Therefore, a fixed concentration method is prone to substantial bias arising from the choice of concentration. In contrast, a fixed ratio has less potential for bias because the maximum difference in the reported MICs between different fixed ratios is equal to the relative difference between the ratio values (e.g., maximum 2x between 1:1 and 1:2 ratios), and the reported MIC will always be nonzero. Together these observations indicate that a fixed ratio method will be a more conservative approach and less likely to misidentify a strain/isolate as sensitive. Hence, while it is already standard practice to use the fixed ratio testing method for β-lactam and β-lactamase inhibitor combinations in which both components are active, a fixed ratio method is also preferable for combinations in which the β-lactamase inhibitor has minimal or no direct antibacterial activity (e.g., avibactam and vaborbactam), for which fixed concentration testing methods are currently used as the default.

Overall, these observations reveal ways in which practices can be improved for the development of new β-lactam and β-lactamase inhibitor combinations in the future. However, there is also the implication that the dose selection for approved β-lactam and β-lactamase inhibitor combinations currently in clinical use may have been suboptimal. Independent support for this premise can be found in the recently published systematic review by Assefa et al. [51]. The authors surveyed the nonclinical evidence that has been used to identify PK/PD indices and target values for approved β-lactam and β-lactamase inhibitor combinations. They concluded “The PK/PD index that describes the efficacy of BLIs [β-lactamase inhibitors] and the exposure measure required for their efficacy is variable among inhibitors; as a result, it is difficult to make clear inference on what the optimum index is.” Since nonclinical PK/PD indices are used to directly determine clinical doses, doubt concerning the best indices and target values leads to uncertainty regarding whether the optimal doses have been selected for clinical use.

## 4. Materials and Methods

Hypothetical checkerboard analysis data for an n x n factorial design investigating combinations of a β-lactam with a β-lactamase inhibitor were schematically represented in a two-dimensional grid with log2 scale axes reflecting standard experimental practice (i.e., nominal concentrations of 1, 2, 4, 8, 16, and 32 µg/mL etc.) (Figure 6). The boundary between areas of growth and no growth defines the apparent MIC boundary. Following convention, each derived MIC value for the β-lactam and β-lactamase inhibitor combination treatment is expressed as the concentration of the β -lactam component without explicit reference to the concentration of the β-lactamase inhibitor component. The same representations of an apparent MIC boundary were used to illustrate the outcome of standard susceptibility testing methods employing a fixed β-lactamase inhibitor concentration or a fixed β-lactam:β-lactamase inhibitor ratio (Figure 7).

Isoboles (i.e., response surfaces representing isoeffective points, see [18,52]) for β-lactam and β-lactamase inhibitor combinations were simulated by adapting models used to evaluate synergy [16,19,24,34]. Simulations included additivity and synergy scenarios for Bliss independence, Loewe additivity, inertism, and highest single agent interaction models. In each case, the drug effects for each combination’s components were assumed to follow a sigmoidal E_max_ relationship, and nominal values were arbitrarily chosen for β-lactam and β-lactamase inhibitor concentrations and model parameters (i.e., E, E_0_, EC_50_, E_max_, and gamma). Synergy was implemented as a β-lactamase inhibitor concentration-dependent scaling factor on the β-lactam drug effect. Factors influencing the β-lactamase inhibitor effect were implemented as changes to relevant E_max_ model parameters (i.e., β-lactamase activity: E_0_; β-lactamase inhibitor K_i_: EC_50_; β-lactamase expression: E_0_ and EC_50_; transport: β-lactamase inhibitor concentration). Isoboles are represented on a two-dimensional figure with log2 scale axes. Simplified representations of simulated isoboles defining an apparent MIC boundary as a single curve were also used to illustrate the results of standard susceptibility testing methods with fixed β-lactamase inhibitor concentrations or fixed β-lactam:β-lactamase inhibitor ratios.

## 5. Conclusions

Thought experiments using theoretical checkerboard MIC curves reveal a number of ways in which current common practices in the development of β-lactam/β-lactamase inhibitor combinations can be improved. Because different mechanisms of action may be dominant in different strains/isolates, data should not be pooled indiscriminately, as assessments of PK/PD indices will be influenced by the composition of a pooled dataset. Similarly, universal application of FIC analysis—which assumes mutual exclusivity—is inappropriate. It will not be possible to derive a universal (PK/PD) index target based on β-lactamase inhibitor C_T_ that is common to all β-lactam partners and dosing regimens, and, hence, priority should be given to in vitro and in vivo models that mimic the clinical dosing regimen of the intended β-lactam partner. A fixed ratio susceptibility testing method is less prone to bias for all β-lactam and β-lactamase inhibitor combinations, including those for which fixed concentration testing methods are currently used as the default. However, susceptibility test MIC values will often not reflect the sensitivity of the strain/isolate to the pharmacological effects of the β-lactamase inhibitor. Therefore, use of susceptibility test MICs to normalize PK/PD indices, as recommended by current regulatory guidelines, is inappropriate for the purposes of identifying β-lactamase inhibitor dosing regimens.

## Figures and Tables

**Figure 1 antibiotics-13-00337-f001:**
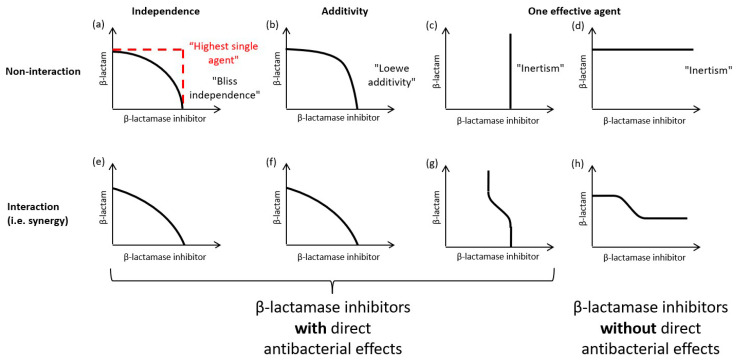
Schematic representation of MIC curves for a β-lactam plus β-lactamase inhibitor combination under selected generalized combination response models. Illustrative MIC curves for independence (**a**,**e**), additivity (**b**,**f**), and one effective agent combination (**c**,**d**,**g**,**h**) response models under scenarios of either no interaction (**a**–**d**) or synergistic interactions (**e**–**h**) represented on a two-dimensional figure with log2 scale axes. Axes represent notional β-lactam (Y axis) and β-lactamase inhibitor (X axis) concentrations in arbitrary units (e.g., µg/mL). Note that antagonism under each model is not shown.

**Figure 2 antibiotics-13-00337-f002:**
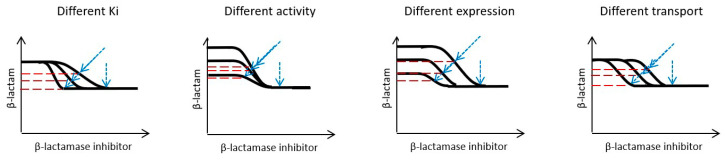
Schematic representation of the qualitative influence of different factors on MIC curves for a β-lactam plus β-lactamase inhibitor combination (a one effective agent synergistic combination). Illustrative MIC curves for one effective agent synergistic combination response models represented on a two-dimensional figure with log2 scale axes. The axes represent notional β-lactam (Y axis) and β-lactamase inhibitor (X axis) concentrations in arbitrary units (e.g., µg/mL). The factors represented include K_i_ for the β-lactamase enzyme, intrinsic β-lactamase enzyme activity, β-lactamase enzyme expression, and β-lactamase inhibitor influx/efflux. Blue arrows represent the standard fixed ratio (diagonal) and fixed concentration (vertical) susceptibility testing methodology. Dashed mauve lines represent the MIC value reported with reference to the β-lactam concentration only.

**Figure 3 antibiotics-13-00337-f003:**
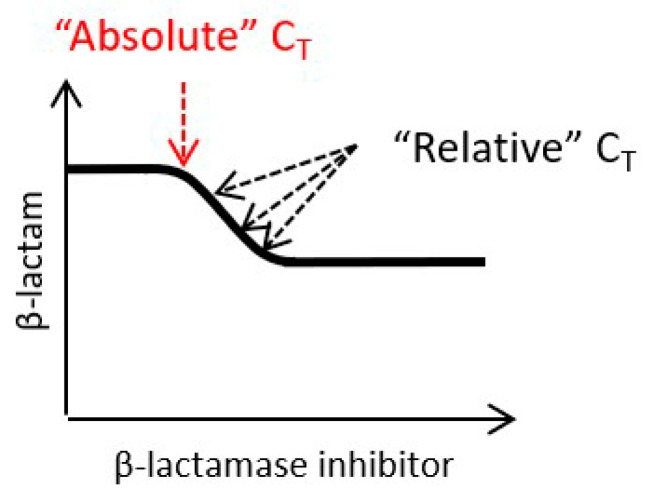
Schematic representation of absolute and relative C_T_ on an MIC curve for a β-lactam plus β-lactamase inhibitor combination (one effective agent synergistic combination). Illustrative MIC curves for one effective agent synergistic combination response models represented on a two-dimensional figure with log2 scale axes. The axes represent notional β-lactam (Y axis) and β-lactamase inhibitor (X axis) concentrations in arbitrary units (e.g., µg/mL). The red arrow represents the absolute C_T_, and the black arrows represent the relative C_T_s.

**Figure 4 antibiotics-13-00337-f004:**
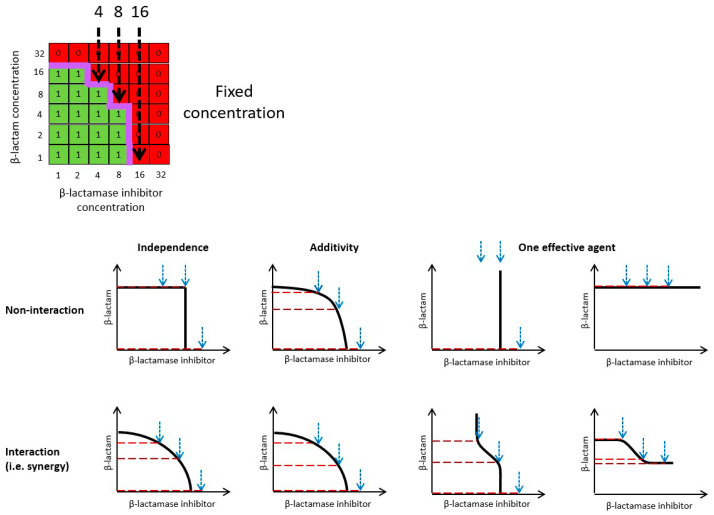
Schematic representation of the fixed concentration susceptibility testing method on hypothetical checkerboard data and representation of MIC curves for a β-lactam plus β-lactamase inhibitor combination. Illustrative MIC curves for independence, additivity, and one effective agent combination response models are represented on a two-dimensional figure with log2 scale axes. The axes represent notional β-lactam (Y axis) and β-lactamase inhibitor (X axis) concentrations in arbitrary units (e.g., µg/mL). The blue arrows represent standard susceptibility testing methodology using a variety of fixed β-lactamase inhibitor concentrations. Dashed mauve lines represent MIC values reported with reference to the β-lactam concentration only.

**Figure 5 antibiotics-13-00337-f005:**
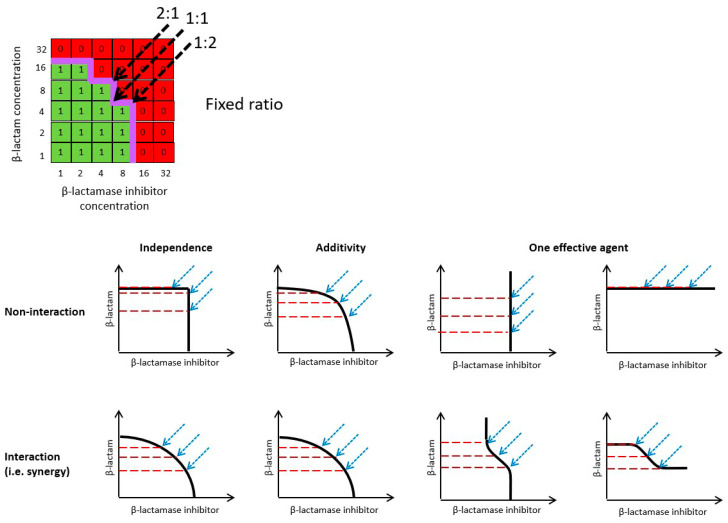
Schematic representation of the fixed ratio susceptibility testing method on hypothetical checkerboard data and representation of MIC curves for a β-lactam plus β-lactamase inhibitor combination. Illustrative MIC curves for independence, additivity, and one effective agent combination response models are represented on a two-dimensional figure with log2 scale axes. Axes represent notional β-lactam (Y axis) and β-lactamase inhibitor (X axis) concentrations in arbitrary units (e.g., µg/mL). The blue arrows represent the standard susceptibility testing methodology using a variety of fixed β-lactam:β-lactamase inhibitor ratios. Dashed mauve lines represent the MIC value reported with reference to the β-lactam concentration only.

**Figure 6 antibiotics-13-00337-f006:**
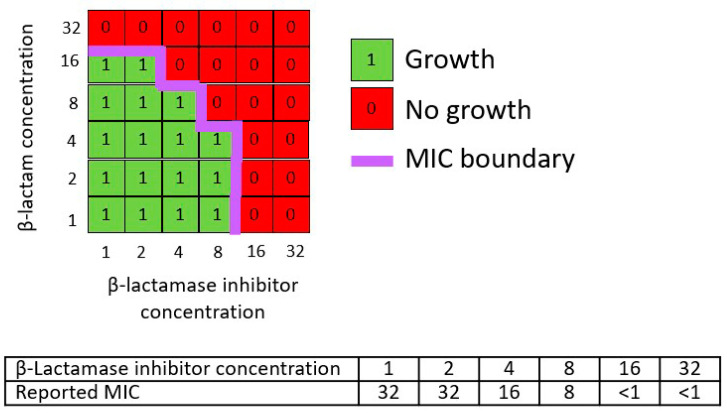
Schematic representation of hypothetical checkerboard data for a β-lactam plus β-lactamase inhibitor combination and reported MIC values. The axes represent notional β-lactam (Y axis) and β-lactamase inhibitor (X axis) concentrations in arbitrary units (e.g., µg/mL).

**Figure 7 antibiotics-13-00337-f007:**
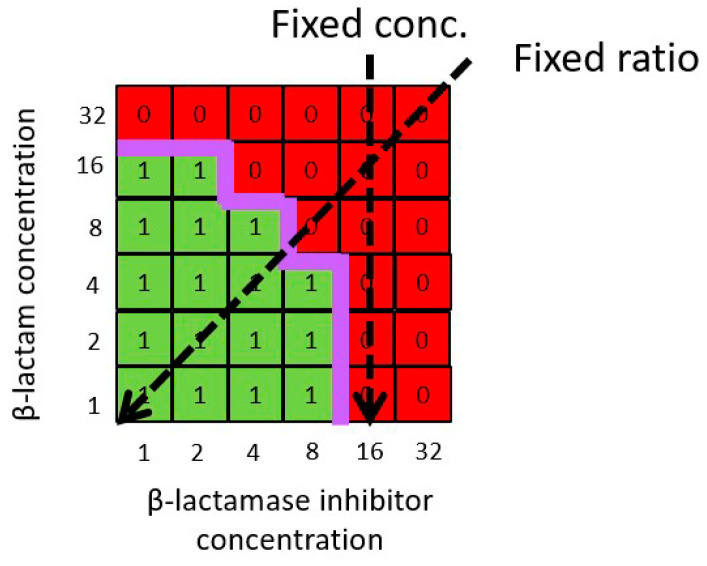
Schematic representation of fixed concentration and fixed ratio susceptibility testing methods on hypothetical checkerboard data for a β-lactam plus β-lactamase inhibitor combination. The axes represent notional β-lactam (Y axis) and β-lactamase inhibitor (X axis) concentrations in arbitrary units (e.g., µg/mL).

## Data Availability

No new data were created or analyzed in this study. Data sharing is not applicable to this article.

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
