# Peer review of "Revisiting the Checkerboard to Inform Development of β-Lactam/β-Lactamase Inhibitor Combinations"

_antibiotics, 2024, doi:10.3390/antibiotics13040337_

Round 1

Reviewer 1 Report

Comments and Suggestions for Authors

Authors describe hypothetical checkerboard analysis of a β-lactam antibiotic with a β-lactamase inhibitor on MIC.

While thought experiments looks impressive, authors should cite clinical studies that used fixed ratio and fixed concentration combination of β-lactam antibiotic with a β-lactamase inhibitor to show the clinical relevance of this work.

How can this theoretical analysis influence clinical practice ? 

Author Response

The author is not aware of any clinical studies for β-lactam/β-lactamase inhibitor combinations that have made comparisons between a fixed ratio and a fixed concentration (e.g. using constant intravenous infusion). This is principally because - for practical reasons - current practice for drug developers is that β-lactam/β-lactamase inhibitor combinations are developed as fixed dose combinations, with choice of doses made early in clinical development. Similarly, as summarized below, all β-lactam/β-lactamase inhibitor combinations in current clinical use employ fixed concentration susceptibility testing methods, although it is noteworthy that some older combinations, e.g. amoxicillin plus clavulanic acid, historically used a fixed ratio testing method but latterly a fixed concentration method has become the accepted norm.

  • Avibactam (ceftazidime) - fixed concentration (4mg/L)
  • Clavulanate (amoxicillin, ticarcillin) - fixed concentration (2mg/L)
  • Durlobactam (sulbactam) - fixed concentration (4mg/L)
  • Relebactam (imipenem+cilastatin) - fixed concentration (4mg/L)
  • Sulbactam (ampicillin, cefoperazone) - fixed concentration (4mg/L)
  • Tazobactam (piperacillin, ceftolozane) - fixed concentration (4mg/L)
  • Vaborbactam (meropenem) - fixed concentration (4mg/L)

Various publications have compared fixed ratio and fixed concentration susceptibility testing methods. However, all comparisons appear to have been based on pre-existing interpretative criteria (breakpoints) or other arbitrarily chosen criteria; the author is not aware of any publications that have rigorously investigated the correlation between different testing methods and clinical efficacy. A comprehensive literature review is considered beyond the scope of this manuscript.

The focus of the manuscript is principally on the drug development process and ways in which practices can be improved for new β-lactam/β-lactamase inhibitor combinations in the future. However, the implication is that dose selection of approved β-lactam/β-lactamase inhibitor combinations currently in clinical use may have been sub-optimal. Unfortunately, the available clinical evidence is largely uninformative. This is principally because dose selection is made early in the clinical development process and there are very few clinical comparisons of efficacy between alternative β-lactam/β-lactamase inhibitor dose combinations. Where clinical dose finding studies have been performed (e.g. Phase 2 studies for relebactam) the study designs were not adequate to differentiate between β-lactamase inhibitor doses on efficacy endpoints. Nevertheless, independent support for the premise that dose selection for approved products has been sub-optimal can be found in the recently published systematic review by Assefa et al (G. M. Assefa, J. A. Roberts, S. A. Mohammed and F. B. Sime. What are the optimal pharmacokinetic/pharmacodynamic targets for beta-lactamase inhibitors? A systematic review. J Antimicrob Chemother. Epublication 2024. DOI: 10.1093/jac/dkae058. The authors surveyed the published non-clinical evidence that had been used to identify PK/PD indices and target values for approved β-lactam/β-lactamase inhibitor combinations, and concluded “The PK/PD index that describes the efficacy of BLIs and the exposure measure required for their efficacy is variable among inhibitors; as a result, it is difficult to make clear inference on what the optimum index is.” Since non-clinical PK/PD indices are used to directly determine clinical doses, uncertainty about the best index and target value leads to doubt that the optimal doses have been selected for clinical use. An additional paragraph to this effect has been added to the discussion and the reference list has been updated (highlighted in yellow in the revised document draft).

Reviewer 2 Report

Comments and Suggestions for Authors

The study refers to an innovative method to assess drug interactions (combination of beta lactamase inhibitors), the study shows us another alternative, however it is necessary to document in a small table in discussion with the sales and disadvantages punctually with other already established methods, since it is an invitro methodology that should be taken to the experimental part to prove its validity.

This information will give us an overview of the applicability and scope of this new strategy, it is not enough with a documentary review, it must be shown how this strategy could be more eligible compared to others or for which specific pharmacological cases it could be applicable.

Author Response

The author acknowledges the reviewer’s comment and agrees with the intention. However, the author has not been able to construct a table that succinctly summarizes the pros and cons of the different approaches in an informative way. Similarly, while it is possible to critically appraise the available data packages for existing β-lactam/β-lactamase inhibitor combinations, the author considers this to beyond the scope of this manuscript. Instead, as described in the response to reviewer #1, an illustration of potential implications for approved β-lactam/β-lactamase inhibitor combinations has been added in an extra paragraph in the discussion. The recently published systematic review by Assefa et al provides independent support for the premise that dose selection for approved combination products has been sub-optimal.